# Detection of a Mixed-Strain Infection with Drug- and Multidrug-Resistant *Mycobacterium avium* Subspecies *hominissuis* in a Dog with Generalized Lymphadenomegaly

**DOI:** 10.3390/antibiotics14040416

**Published:** 2025-04-19

**Authors:** Cinzia Marianelli, Angelo Leonori, Romana Stecco, Carlo Giannantoni

**Affiliations:** 1Department of Food Safety, Nutrition and Veterinary Public Health, Istituto Superiore di Sanità, 00161 Rome, Italy; 2Ambulatorio Veterinario Leonori, 02032 Fara in Sabina, Italy; studioveterinarioleonori@gmail.com (A.L.); romanastecco@gmail.com (R.S.); 3Centro Veterinario Reatino, 02100 Rieti, Italy; ambvetrieti@gmail.com

**Keywords:** *Mycobacterium avium* subspecies *hominissuis*, nontuberculous mycobacteria infection, mixed-strain infection, MIRU-VNTR typing, SNPs typing, *canis lupus*

## Abstract

**Background** Members of the *Mycobacterium avium* complex (MAC) have been documented to cause severe and disseminated infections in dogs, although such cases are sporadically reported. In this study, a comprehensive account of a rare case of generalised lymphadenomegaly caused by a mixed-strain infection with drug- and multidrug-resistant *Mycobacterium avium* subspecies *hominissuis* (Mah) in a Maremma sheepdog is presented. **Methods** Laboratory investigations, as well as the monitoring of the clinical signs displayed by the animal, were conducted throughout the course of a two-year drug therapy (based on rifampicin, azithromycin, and ciprofloxacin) and a two-year post-treatment follow-up period, until the death of the dog. Laboratory examinations included both solid and broth cultures from fine-needle aspiration samples of lymph nodes, molecular typing by 8-locus MIRUVNTR analysis and SNPs typing of five genetic regions (*gyrB*, *rpsA*, *3′hsp65*, ITS and *rpoB*), and drug susceptibility testing towards seven antimycobacterial drugs. **Results** The results indicated the presence of two distinct genotypes of Mah, which exhibited different phenotypic characteristics, such as different drug susceptibility profiles and growth abilities in broth and solid media, suggesting a mixed-strain infection. Resistances to ethambutol alone, to ethambutol and clarithromycin, and to ethambutol, clarithromycin, rifampicin, and doxycycline were detected over the study. **Conclusions** Although the Mah strains isolated during the course of therapy showed sensitivity to the regiment, the complete eradication of the infection was never achieved. It has been hypothesised that the presence of drug-resistant and multidrug-resistant Mah strains in the animal may have been established at the onset of the infection or soon thereafter. The exposure to therapy has been suggested as a potential factor that could have favoured the growth of resistant strains, thereby rendering the therapy ineffective. The implications that the distinct phenotypic and genotypic profiles of Mah described here may have had for disease dynamics and control are discussed.

## 1. Introduction

*Mycobacterium avium* complex (MAC), which consists of multiple nontuberculous mycobacteria (NTM) species, is frequently isolated in human infections, particularly in cases of NTM lung disease [1]. To date, MAC comprises twelve species; the most clinically relevant are *M. avium*, *M. intracellulare*, and *M. chimaera* [2]. *Mycobacterium avium* is currently divided into four subspecies, namely *M. avium* subsp. *avium* (Maa), *M. avium* subsp. *silvaticum* (Mas), *M. avium* subsp. *hominissuis* (Mah), and *M. avium* subsp. *paratuberculosis* (Map). Each of these subspecies possesses specific characteristics in terms of pathogenicity and host spectrum [3]. MAC comprises environmental bacteria that can cause opportunistic and severe infections in both humans and animals. Of these, Mah, a ubiquitous environmental saprophyte, causes chronic infections in a wide range of hosts, including dogs [4,5,6,7].

Dogs possess a substantial innate capacity to resist MAC infections. However, severe systemic or disseminated MAC infections have been documented particularly in two breeds: Miniature Schnauzer and Basset Hound. In these breeds, a genetic predisposition to MAC infections has been documented [4,7,8,9]. On rare occasions, similar cases have also been documented in a limited number of other breeds [5,10,11]. Disseminated MAC infections have been reported to be associated with a poor prognosis [12,13]. In veterinary medicine, the treatment of such infections is controversial, as there are no approved drugs for animals and MAC responds poorly to treatment. Drugs used for MAC infections in humans are generally used [14]. A long-course therapy, with an initial phase of two or three drugs (rifampicin, fluoroquinolone, clarithromycin/azithromycin) for two months, followed by two drugs (rifampicin and either a fluoroquinolone or clarithromycin/azithromycin) for a further four to six months, may be prescribed [14]. It is important to note that the treatment of systemic mycobacteriosis is generally not effective, and therefore it is not recommended in dogs [4,11].

The progression and outcome of a mycobacterial infection may be further complicated when different mycobacterial species or genetically distinct strains of the same pathogenic species concurrently infect a single host [15]. These mixed infections, which may involve members of the *Mycobacterial tuberculosis* complex or NTM, may occur in humans, as well as in a variety of animal species. The majority of studies have been conducted in humans, with a predominant focus on *M. tuberculosis*. In contrast, research examining the prevalence and impact of mixed mycobacterial infections in veterinary medicine is limited. [15]. The presence of a mixed mycobacterial infection can be difficult to detect, which can result in complications in the diagnostic process, treatment regimen selection, and subsequent course of the disease.

The present study aims at describing the diagnosis and the long-course treatment, as well as the clinical outcomes, of a rare case of generalised mycobacteriosis in a dog, caused by a mixed-strain infection with drug- and multidrug-resistant Mah. Furthermore, the study explores the implications that the distinct phenotypic and genotypic profiles described here may have had in the disease dynamic and control.

## 2. Case Description

### 2.1. Diagnosis of a MAC Infection and Administration of Therapy

In April 2018, a female Maremma sheepdog of approximately two years of age was retrieved from the street. The dog was emaciated and starving and presented patches of hair loss on her body. The dog’s weight was 19 kg (Figure 1A). A series of tests was conducted to ascertain the absence of several infectious agents, including scabies, Leishmania, Ehrlichia, and Rickettsia. The dog was provided with nourishment and care by the rescuer. The haloes were cleaned on a regular basis with Marseille soap enriched with sulphur.

Two months later, the dog exhibited signs of weight gain and hair regrowth and thus was transferred to a dog shelter for potential adoption. Following a further four months, the animal was adopted by an individual who had retrieved it from the street and relocated it to the family residence. The animal appeared weary and still underweight. A comprehensive veterinary examination was conducted, which revealed the presence of splenomegaly and systemic lymphadenomegaly, as indicated by ultrasound imaging. For further analysis, fine-needle aspiration (FNA) samples of the spleen as well as mesenteric and pre-scapular lymph nodes were collected for cytological examination.

In the May–Grünwald–Giemsa stain, a mixed population of small and medium-sized lymphocytes and numerous macrophages was observed, with negatively stained bacilli present inside and outside of macrophages (Figure 2). The ultrasound and cytological features were consistent with a granulomatous disease caused by atypical mycobacteria, probably related to MAC, and affecting the spleen and mesenteric and superficial lymph nodes. To confirm the diagnosis, FNA samples of prescapular lymph nodes were taken for bacterial culture and molecular identification. While awaiting results of the diagnostic investigations, the dog was treated orally with rifampicin (300 mg/12 h), azithromycin (160 mg/day) and enrofloxacin (15 mg/day) in early January 2019. Omeprazole (20 mg/12 h) was introduced as a gastroprotective agent. In addition to the prescribed treatment, the dog owner administered a Bach Flower-based mixture comprising ild rose, sweet chestnut, Scleranthus, gorse, impatiens, star of Bethlehem, crab apple, and cherry plum.

The FNA samples were also subjected to microbiological and molecular analyses in the laboratories of the Istituto Superiore di Sanità (ISS).

### 2.2. Microbiological and Molecular Investigations

For cultural examination, an FNA sample was inoculated into Middlebrook 7H9 broth (Biolife, Milan, Italy) with 10% oleic acid–albumin–dextrose complex (OADC) (Becton, Dickinson and Company, NJ, USA). Meanwhile, an additional FNA sample was inoculated onto Middlebrook 7H11 + 10% OADC agar plates. The samples were then incubated at 37 °C under aerobic condition. Bacterial growth was monitored at two-day intervals. Following a 14-day incubation period, both broth and solid cultures exhibited signs of bacterial growth. The extraction of DNA was carried out on both primary cultures using a commercial kit (InstaGene Matrix; Bio-Rad Laboratories, Milan, Italy), in accordance with the manufacturer’s instructions. Two µL of DNA from each sample was used as a template for the insertion sequence (IS)-based PCR tests developed at the ISS, with the objective of detecting and differentiating *M. avium* members (Table 1). Reactions were performed in a total volume of 50 µL using MyTaq Red DNA Polymerase (Bioline, London, UK) in accordance with the manufacturer’s instructions and the following cycling conditions: denaturation at 94 °C for 5 min, followed by 35 cycles at 94 °C for 30 s, 57 °C for 30 s, and 72 °C for 60 s, and a final extension cycle at 72 °C for 10 min. Field isolates of *M. avium* from our collection were used as controls. The amplification of 356 bp-long and 283 bp-long fragments in all samples suggested the identification of a *M. avium* pathogen (356 bp fragment) and the differentiation of the pathogen as Mah (283 bp fragment).

### 2.3. MIRU-VNTR Analysis

In order to confirm the diagnosis of a Mah infection, the molecular investigation was extended using the 8-locus MIRU-VNTR typing method, a minisatellite typing system previously delineated by Thibault et al. [16]. This approach involved the PCR amplification of DNA at eight loci: TR32, TR392, TRX3, TR25, TR3, TR7, TR10, and TR47. Reactions were performed in a total volume of 50 μL using MyTaq Red DNA Polymerase (Bioline) and following the protocol previously described by Marianelli et al. [5]. To estimate exactly the amplicon sizes and thus determine the number of repeated units for each locus, all PCR products were purified and then sequenced using the same PCR primers. The Codon Code Aligner software 9.0.1 (CodonCode Corporation, MA, USA) was used for forward and reverse sequence assembly. The MIRU-VNTR profile was determined using the table showing the theoretical size of PCR products available at the MAC-INMV-SSR database website “http://mac-inmv.tours.inra.fr/index.php?p=fa_download (accessed on 10 January 2025)” [17]. The Mah isolates from both the solid and broth primary cultures exhibited identical MIRU-VNTR profiles: 2, 2, 2, 2, 1, 1, 2, 8. This MIRU-VNTR profile was subsequently run against the freely accessible MAC-INMV-SSR database “http://mac-inmv.tours.inra.fr/index.php (accessed on 10 January 2025)” in order to identify the pathogen at the sub-species level. The profile perfectly matched Mah INMV 92 in the MAC-INMV-SSR database, thus confirming the diagnosis of a Mah infection. The Mah pathogens with MIRU-VNTR genotype 2, 2, 2, 2, 1, 1, 2, 8 from broth and solid cultures were thus designated as ‘Mah-Dec2018’.

### 2.4. Drug Susceptibility Testing

In order to ascertain that the dog was receiving an appropriate therapy, the broth and solid primary cultures were subcultured in broth and then assessed for drug susceptibility using the resazurin microtitre assay (REMA), in accordance with the protocol that had been previously described by Armas et al. [11]. The assay was conducted in triplicate in 96-well plates. Briefly, 100 mL of Middlebrook 7H9 broth with 10% OADC enrichment were dispensed into each well. Bacterial suspensions were prepared by adjusting them to 1.0 McFarland and subsequently diluting them 1:20 in the same medium. The inoculum volume was 100 mL. The antituberculosis drugs were added subsequently. A growth control containing no drug and a sterile control without inoculum were also included in the test. The susceptibility of the Mah-Dec2018 isolates to rifampicin, the macrolide clarithromycin (which is the recommended class antibiotic for in vitro testing due to technical issues with azithromycin), and ciprofloxacin (which is the active metabolite of enrofloxacin) was tested (Sigma-Aldrich, Gillingham, UK). Sensitivity to four additional drugs was also tested: streptomycin, amikacin, doxycycline, and ethambutol (Sigma-Aldrich, UK). The minimal inhibitory concentration (MIC) breakpoints for each antimicrobial agent previously described by Marianelli et al. [5] were utilised to ascertain drug sensibility/resistance. The antibiotic susceptibility testing was performed three times, independently, to ensure the reliability of the results.

The drug susceptibility profiles of samples from broth and solid cultures were found to be identical. The Mah-Dec2018 pathogens exhibited sensitivity to rifampicin, clarithromycin, ciprofloxacin, streptomycin, amikacin, and doxycycline but resistance to ethambutol. The resazurin blue dye was reduced to pink at the resistant breakpoints for ethambutol (R, ≥ 8.0 μg mL^−1^) (Figure 3A, columns 9–12, lines D–F). Consequently, the Mah-Dec2018 pathogens demonstrated resistance to ethambutol exclusively. The therapy based on rifampicin, azithromycin, and enrofloxacin was confirmed.

### 2.5. Modification of the Treatment Plan and Improvement in Health Clinical Conditions

Fifteen days following the commencement of the therapeutic regimen, the canine subject exhibited indications of lethargy, despondency, and anorexia. Additionally, the animal displayed joint discomfort, initially in the hind legs and subsequently in the front legs, which resulted in marked difficulty in ambulation. The radiological examination, however, did not identify any lesions. The antibiotic enrofloxacin was identified as a possible cause of lameness in the animal and thus suspended from the treatment plan. The regimen was augmented with a non-steroidal anti-inflammatory drug (Celecoxib 150 mg/day) for 15 days to manage any residual pain. As the clinical signs persisted, the owner, administered Enteromicro Complex (2 cp/12 h)—a complementary feed containing probiotics and prebiotics—and glutathione (350 mg/day) orally for 20 days in conjunction with the pharmacological therapy. Subsequently, the owner administered Lynphomyosot Hell vials (1 vial/day), a homoeopathic remedy that promotes natural lymphatic drainage, for a period of 10 days. This was followed by a further cycle of administration of glutathione and Lynphomyosot, with Enteromicro Complex continued throughout.

Three months following the commencement of the therapeutic intervention, a marked improvement in the animal’s clinical condition was observed, as evidenced by a reduction in lymphadenomegaly. This improvement persisted, and a substantial recovery was documented after a further three months. The dog exhibited signs of improved appetite and weight gain (Figure 1B). The systemic lymphadenomegaly showed signs of reduction, with the exception of the mesenteric lymph nodes, which continued to demonstrate signs of enlargement on ultrasound imaging. The normalization of the lymphocyte count and haematocrit levels further corroborated a favourable prognosis. Nonetheless, the cytological examination of FNA samples of prescapular lymph nodes continued to reveal the diffuse presence of acid-fast bacilli within and outside of the macrophages. In an effort to eradicate or mitigate the mycobacterial infection, the therapeutic approach was adapted. The dosages of rifampicin and azithromycin were increased to 450 mg/12 h and 250 mg/day, respectively; enrofloxacin (15 mg/day) was added, and omeprazole (20 mg/12 h) was continued.

Three months after the administration of the new treatment, enrofloxacin was dismissed and FNA samples of prescapular lymph nodes were collected for microbiological and molecular analysis.

### 2.6. Further Improvement in Overall Health Status While Mah Infection Was Still Present

The cultural and molecular investigations were conducted as described above. The IS-based PCR assays and 8-locus MIRU-VNTR typing confirmed the pathogen as Mah, here named ‘Mah-Nov2019’, which showed an identical MIRU-VNTR profile to Mah-Dec2018 (profile 2, 2, 2, 2, 1, 1, 2, 8). Furthermore, the drug susceptibility testing was repeated, and samples from both and solid cultures confirmed the sensibility to all tested drugs but ethambutol, similar to Mah-Dec2018 (Figure 3A). The therapeutic regimen, comprising rifampicin, azithromycin, and omeprazole, was confirmed.

In June 2020, following a period of 18 months from the initial diagnosis and commencement of drug therapy, the dog, with a recorded weight of 42 kg, appeared to be in good health, exhibiting only a mild form of incontinence on occasion. Ultrasound imaging revealed that the palpable and accessible lymph nodes had returned to almost their original dimensions, with the exception of the mesenteric lymph nodes, which continued to exhibit signs of enlargement. Furthermore, mild splenomegaly was still observed. Once more, FNA samples were collected for further analysis, specifically for cytological and microbiological, as well as molecular investigations. The cytological analysis revealed the presence of lymphocytes of both small and medium size, in addition to a modest number of macrophages and a few rod-shaped elements attributable to mycobacteria. However, the number of mycobacteria observed in this examination was lower than in previous cytological analyses. The proposed therapeutic regimen was subsequently confirmed.

### 2.7. Interruption of Drug Therapy Due to the Occurrence of Medical Complications

In early February 2021, following a period of over two years of uninterrupted therapy based on rifampicin, azithromycin, and omeprazole, the dog exhibited an escalation in the severity of incontinence, progressing from a mild to a moderate state. Concurrently, the discomfort in the limbs and the impediments to mobility reemerged. Comprehensive blood and urine analyses were conducted, which revealed a significant hepatic dysfunction. Consequently, the administration of antibiotic therapy was promptly discontinued in late February 2021.

Following the cessation of the therapeutic regimen, there was a marked improvement in the dog’s health status. One month later, the pain in the legs and the limp had resolved; the blood analysis demonstrated complete restoration of liver function. FNA samples of prescapular lymph nodes were collected for microbiological and molecular investigations.

### 2.8. Isolation of a Mah Pathogen with New Phenotypic and Genotypic Characteristics

The cultural and molecular investigations were conducted in accordance with the aforementioned methodology. In contrast with previous observations, bacterial growth was detected exclusively on the agar plate spread with the FNA sample. No growth was observed in the inoculated broth culture. The extraction of DNA was performed from the biomass harvested from the agar plate. The IS-based PCR assays confirmed the presence of a Mah pathogen. However, the 8-locus MIRU-VNTR typing identified a new MIRU-VNTR profile, which carried a polymorphism in the second locus investigated (locus TR392) when compared to the previous Mah-Dec2018 and Mah-Nov2019 isolates: five instead of two, as shown in Figure 4. The newly identified Mah pathogen with MIRU-VNTR genotype 2, 5, 2, 2, 1, 1, 2, 8—here designated ‘Mah-Apr2021’—exhibited a perfect match with the reference Mah INMV 107 in the MAC-INMV-SSR database. The biomass harvested from the agar plate was subcultured in Middlebrook 7H9 broth before performing REMA, as described above. The Mah-Apr2021 pathogen exhibited resistance to both clarithromycin (R, ≥ 64.0 μg mL^−1^) (Figure 3B, columns 1–3, lines A–C) and ethambutol (R, ≥ 8.0 μg mL^−1^) (Figure 3B, columns 9–12, lines D–F).

### 2.9. Reduction in Lymphadenomegaly and Isolation of Mah Pathogens with Divergent Drug Susceptibility Profiles

In April 2022, twelve months after the interruption of the therapy, the dog exhibited no signs of illness with the exception of the mesenteric lymph nodes, which continued to exhibit signs of enlargement. FNA samples of popliteal lymph nodes were collected and sent to ISS for microbiological and molecular investigations, as previously outlined. Bacterial growth was observed on the agar plates and, to a lesser extent, in the broth. The extraction of DNA was performed from both broth and solid cultures and then underwent molecular analysis. Both IS-based PCR assays and 8-locus MIRU-VNTR typing confirmed the pathogen as Mah, here named ‘Mah-Apr2022’, which showed an identical MIRU-VNTR profile to the previous Mah-Apr2021 isolate (profile 2, 5, 2, 2, 1, 1, 2, 8). Both the primary cultures were then subjected to susceptibility testing by REMA, as described above.

Discrepancies in drug susceptibility were observed. Mah-Apr2022 from the broth culture showed susceptibility to all tested drugs except ethambutol, similar to Mah-Dec2018 and Mah-Nov2019 (Figure 3A). Conversely, Mah-Apr2022 from the solid culture showed resistance not only to clarithromycin (R, ≥ 64.0 μg mL^−1^) (Figure 3C, columns 1–3, lines A–C) and ethambutol (R, ≥ 8.0 μg mL^−1^) (Figure 3C, columns 9–12, lines D–F) but also to rifampicin (R, ≥ 2.0 μg mL^−1^) (Figure 3C, columns 1–2, lines D–F) and doxycycline (R, ≥ 16.0 μg mL^−1^) (Figure 3C, columns 6–8, lines D–F), suggesting heterogeneity in drug susceptibility.

### 2.10. Recurrence of the Disease and Subsequent Demise of the Animal

In November 2022, twenty months after the cessation of the therapeutic regimen, the dog showed a recurrence of the disease. FNA samples of spleen, and prescapular lymph nodes were collected for microbiological and molecular investigations. The analyses of both broth and agar cultures confirmed the presence of Mah in all samples, here designed ‘Mah-Nov2022’, with an 8-locus MIRU-VNTR profile of 2, 5, 2, 2, 1, 1, 2, 8, which was found to be identical to that of Mah-Apr2021 and Mah-Apr2022. Subcultures from both broth and agar cultures showed resistance to ethambutol only, similar to Mah-Dec2018 and Mah-Nov2019 (Figure 3A).

A month later, during the course of the investigation, the dog displayed symptoms of respiratory distress, which rapidly escalated. The dog succumbed to pulmonary haemorrhage within a few days. To the best of our knowledge, this is the first report detailing the survival of a dog with a diagnosis of generalised mycobacteriosis for a period exceeding four years from the initial diagnosis. Notably, this survival was achieved following a continuous treatment regimen involving a two-year course of medication. A chronological overview of the most significant events is presented in Figure 5.

### 2.11. SNP Typing of All the Mah Isolates

In order to enhance the molecular characterisation of the Mah isolates, the single nucleotide polymorphisms (SNPs) typing procedure was conducted. The following genetic regions were amplified and sequenced: the genes *gyrB*—encoding the DNA gyrase B (PCR fragment of 353 bp)—and *rpsA*—encoding the ribosomal protein (PCR fragment of 933 bp)—previously described by Armas and collaborators [11], together with the more variable 3′ region of the *hsp65* gene (3′*hsp65*) (PCR fragment of 1059 bp) described by Turenne et al. [18] and the complete 16S-23S rDNA internal transcribed spacer (ITS) (PCR fragment of about 600 bp in length) described by Frothingham and Wilson [19]. Furthermore, a region of the *rpoB* gene—encoding the beta subunit of the DNA-directed RNA polymerase—was identified in this study for nucleotide polymorphisms through multiple sequence alignment of *rpoB* nucleotide sequences from MAC reference strains available at NCBI. Two conserved regions flanking the polymorphic site were selected for the design of forward and reverse primers capable of amplifying a 499 bp fragment of the *rpoB* gene. The complete set of primers utilised for both PCR amplifications and sequencing can be found in Table 2. Reactions were performed in a total volume of 50 µL using MyTaq Red DNA Polymerase (Bioline) in accordance with the manufacturer’s instructions. The cycling conditions for all PCR amplifications were as follows: denaturation at 94 °C for 5 min, followed by 35 cycles at 94 °C for 20 s, 57 °C for 30 s, and 72 °C for 60 s, and a final extension cycle at 72 °C for 7 min. All PCR products were purified and then sequenced using the same PCR primers. The results were analysed using ABI Prism SeqScape software, version 2.0 (Applied Biosystems, Thermo Fisher Scientific, MA, USA). The consensus sequences generated by aligning forward and reverse reads were compared to each other and to the reference strain Mah MAC109 (complete genome available in GenBank, sequence ID CP029332.1), using the Blast tool “http://blast.ncbi.nlm.nih.gov/Blast.cgi (accessed on 16 October 2024)” to detect nucleotide discrepancy. In order to confirm the results, DNA extraction, PCR amplifications, and nucleotide sequencing of the amplicons were repeated twice, independently.

The nucleotide sequences of *gyrB*, *rpsA*, ITS, and *rpoB* amplicons exhibited 100% identity when all Mah pathogens—Mah-Dec2018, Mah-Nov2019, Mah-Apr2021, Mah-Apr2022, and Mah-Nov2022—were compared to each other, and 98–100% identity when compared to the reference MAC 109. Conversely, nucleotide sequence discrepancies were observed in the 3′*hsp65* gene at five sites, which resulted in changes to the nucleotide sequence but not to the amino acid sequence. The Mah pathogens isolated in 2018 and 2019 showed an identical 3′*hsp65* nucleotide sequence to the reference MAC 109. Conversely, the Mah pathogens isolated in 2021 and 2022 carried five silent mutations at codons 376, 406, 423, 424, and 512 (Table 3). The MIRU-VNTR polymorphism at locus TR392 and the five silent mutations suggested the idea of an MSI formed by two distinct genotypes: Mah-Dec 2018 and Mah-Nov2019 on the one hand (genotype A), and Mah-Apr2021, Mah-Apr2022, and Mah-Nov2022 on the other (genotype B).

### 2.12. Clustering of the Mah Isolates

According to the drug susceptibility phenotype, the Mah isolates could be divided into three distinct groups (see Figure 3). These groups were as follows: ethambutol-resistant profile (panel A), clarithromycin- and ethambutol-resistant profile (panel B), and clarithromycin-, rifampicin-, doxycycline-, and ethambutol-resistant profile. By combining these three different phenotypes with the two distinct genotypes, the Mah isolates were found to group into four distinct clusters, as shown in Figure 5: genotype A showing ethambutol resistance (Mah-Dec2018 and Mah-Nov2019), genotype B exhibiting clarithromycin and ethambutol resistance (Mah-Apr2021), genotype B showing ethambutol resistance (Mah-Apr2022 and Mah-Nov2022), and genotype B exhibiting clarithromycin, rifampicin, doxycycline, and ethambutol resistance (Mah-Apr2022). All the above findings suggested an MSI of four distinct Mah strains.

## 3. Discussion

Mixed mycobacterial infections have been documented in both human and animal populations, though the majority of extant research has focused on *Mycobacterium tuberculosis* [15]. The development of molecular typing methods, including whole-genome sequencing (WGS), MIRU-VNTR, and other PCR-based genotyping techniques, has led to significant advancements in the field of pathogen detection. These techniques have enabled the revelation of mixed infections with genetically distinct *M. tuberculosis* strains within a single host [20,21,22] and even within a single organ [23]. Mixed infections have been documented in different settings [24,25,26].

As stated in the research by Cohen et al. [24] and Muwonge et al. [27], mixed infections with *M. tuberculosis* may occur as a result of either within-host strain evolution following a single infection event—known as microevolution or clonal heterogeneity—or sequential (or simultaneous) exposure to more than one strain of *M. tuberculosis*—known as mixed-strain infection (MSI). Microevolution is characterised by the sporadic occurrence of sequential SNPs through mutations, resulting in genetically distinct isolates [28,29,30]; it is contingent on within-host processes [25]. MSI, on the other hand, enables a host to acquire entirely new *M. tuberculosis* genomes [25,31]. The acquisition rate of MSIs is dependent on the force of infection in the community and the diversity of strains in that environment [25].

Few studies have documented mixed NTM infections in animals. Byrne et al. [32] studied 14 Map-infected, high-shedding dairy cattle and recorded the presence of MSIs and microevolution in almost all animals using WGS, multi-locus short sequence repeat (SSR)- and SNP-based analyses. Mixed Map infections have also been identified both in domestic ruminant species using multi-locus SSR typing [33] and 8-locus MIRU-VNTR [34,35], as well as in wildlife [35]. In addition to Map, other NTM have been shown to cause MSIs. Mixed Maa infections were identified in domestic chickens using IS901-RFLP [36]. Furthermore, mixed infections with Maa have been documented in pigs using 8-locus MIRU-VNTR [37] or IS1245-RFLP [38] and in birds [39] using WGS. Marianelli et al. [5] reported the isolation of two distinct genotypes of Mah from a dog with generalised mycobacteriosis using SNP typing. Silva-Pereira et al. [40] detected an *M. pinnipedii* MSI in a South American sea lion (*Otaria flavescens*) using WGS. In addition to MSIs, microevolution events involving Mah have been documented in bongo antelopes using 8-locus MIRU-VNTR and IS1245-RFPL [41]. Furthermore, microevolution of *M. caprae* has been identified in various animals, including goats, cattle, sheep, and wild boar by 8-locus MIRU-VNTR [42]. In this study, we documented a case of MSI with at least four distinct Mah strains in a dog, as evidenced by the identification of a polymorphism at the locus TR392 and five SNPs within the 3′*hsp65* gene, in conjunction with phenotypic heterogeneity in drug susceptibility.

Although different genotyping methods and various molecular approaches can be employed for the detection of mixed infections in clinical samples, WGS and MIRU-VNTR techniques are the most sensitive, exhibiting higher discriminatory power than many other widely used techniques [43]. The MIRU-VNTR typing method, which differentiates *M. tuberculosis* strains according to minisatellite copy number variation (CNV) at a limited number of specific loci, is the most widely used technique for detecting mixed infections [44,45], even though WGS offers higher resolution and can distinguish highly related but genetically distinct strains and estimate the true extent of MSIs [25,46]. In addition to WGS, metagenomic tools have also demonstrated high resolution in the identification of MSIs, even in the analysis of ancient DNA found in mummified bodies [47,48]. While WGS provides the highest resolution for genome-based species identification and can provide insight into the antimicrobial resistance and virulence potential of a single microbiological isolate, metagenomics allows the analysis of DNA segments from multiple microorganisms within a community. The combined application of both WGS and metagenomics has the potential to generate additive or synergistic information, which is critical for microbiological diagnosis, patient management, infection control, and pathogen surveillance. However, to date, only a limited number of studies have successfully documented the integrated use of both technologies [49].

According Chindelevitch et al. [50], a mixed *M. tuberculosis* infection is typically characterised by the presence of CNVs resulting from either microevolution where multiple CNVs are found in only one locus, or MSI where two or more loci have multiple CNVs. However, it is challenging to establish definitive distinctions between clonal heterogeneity and MSIs based solely on discrepancies in MIRU-VNTR patterns [24,25,51]. In this study, we used two typing methods to characterised isolates, namely SNP typing of five genetic regions—namely the genes *gyrB*, *rpsA* and *rpoB*, the more variable 3′ region of the *hsp65* gene, and the complete 16S-23S rDNA ITS—and MIRU-VNTR analysis of eight loci. We found CNVs at one locus (the TR392 locus) and five SNPs in the 3′*hsp65* gene (five silent mutations at codons 376, 406, 423, 424, and 512). These genetic polymorphisms grouped all Mah pathogens into two groups—Mah-Dec2018 and Mah-Nov2019 on the one hand, and Mah-Apr2021, Mah-Apr2022, and Mah-Nov2022 on the other. Despite the presence of CNVs at a single locus, the additional detection of five mutations in the 3′*hsp65* gene indicated the occurrence of an MSI with two distinct Mah strains. It is plausible that a higher number of loci included in the MIRU-VNTR analysis (such as the 12 or 24 MIRU-VNTR loci used for *M. tuberculosis* typing) would have detected additional CNVs.

In addition to genetic polymorphisms, the two distinct Mah strains exhibited distinct phenotypic characteristics, including different drug susceptibility profiles and growth abilities in broth and solid media. This further corroborated the hypothesis of MSI. It has been reported that the different phenotypic characteristics of the strains in a mixed infection have the capacity to modify disease dynamics and control thereby complicating the diagnostic process and rendering the standard treatment ineffective [25,32,44,50,52,53]. Furthermore, MSIs can have consequences such as altering the immune response—possibly due to antigenic differences between the infecting strains—and increasing the patient’s susceptibility to reinfection [25,32].

The Mah pathogens isolated when the dog was on the three-drug regimen showed resistance to ethambutol only and therefore susceptibility to the three-based therapy (rifampicin, azithromycin, and ciprofloxacin). Despite this, the two-year course of medication failed to eliminate the Mah infection, as demonstrated by cytological and microbiological investigations. Following the interruption of the therapeutic regimen, a novel Mah genotype was identified, characterised by polymorphisms in the TR392 locus and 3′*hsp65* gene. This new genotype demonstrated not only resistance to ethambutol but also to clarithromycin. Twelve months after the completion of therapy, resistance to both rifampicin and doxycycline was further detected. Given the ineffectiveness of the therapy, it is plausible that the drug-resistant and multidrug-resistant Mah strains might have been present in the animal since the onset of the infection, or soon thereafter. Exposure to rifampicin-, azithromycin-, and ciprofloxacin-based therapy might have favoured the growth of resistant strains, thus rendering therapy ineffective. It is known that, within the context of a mixed infection, drug susceptibility testing may not detect minority drug-resistant strains. This failure to detect such minority strains can result in adverse treatment outcomes due to underlying resistance, as well as an increased risk of acquiring resistance when using the standard treatment regimen [45,54]. This has a detrimental effect on the clinical outcomes of patients [25]. This potential scenario could have occurred in the present case. As with drug susceptibility testing, genotyping methods are only able to identify the dominant genotype presents in a mixed infection. This means that minor genotypes are not detected. Using our diagnostic approach, we were able to detect two different genotypes, which were undoubtedly the predominant ones, whilst the minority ones remained undiscovered. The identification of three distinct drug susceptibility phenotypes and two separate genotypes indicated that the MSI under investigation here involved at least four distinct Mah strains. It can be hypothesised that the implementation of a more sensitive technique, such as WGS, could potentially uncover a greater number of Mah strains.

Following the termination of therapy, resistance to ethambutol alone reemerged; this remained the only identifiable phenotype twenty months later. It is known that MAC may infect, replicate, and survive within macrophages similar to *M. tuberculosis*. MAC infections are indeed associated with relapses and the development of chronicity [55]. The ability of bacteria to survive without undergoing growth within host cells poses significant challenges in the identification of drugs that can interfere with the survival of non-proliferating bacteria [56]. It is therefore conceivable that the Mah strains exhibiting resistance to ethambutol alone (and thus sensitivity to the three-drug regimen) could have survived the treatment by hiding within macrophages in a non-replicative state. Subsequent to the cessation of the therapeutic intervention, these susceptible strains could have initiated a process of proliferation that could have resulted in their increased prevalence over others. This potential scenario may provide a rationale for our capacity to redetect sensitive strains to the regimen (and resistant to ethambutol only) at twelve and twenty months following the termination of the therapy.

A salient characteristic of NTM is the high level of drug resistance they exhibit. This factor renders the treatment of NTM significantly more challenging in comparison to that of *Mycobacterium tuberculosis* complex. A number of studies have documented the isolation of drug-resistant and multidrug-resistant MAC. Resistance to linezolid and moxifloxacin has been recently documented in clinical isolates of *M. avium* [57]. In addition, further resistances to isoniazid, rifampin, doxycycline, ethionamide, ciprofloxacin, ethambutol, streptomycin, cotrimoxazole, and clarithromycin have also been recorded in *M. avium* clinical isolates [58,59]. Moreover, *M. avium* generally exhibits higher levels of resistance to antibiotics in comparison to the other MAC members, such as *M. intracellulare* and *M. chimaera* [57,58,59]. A recent study has described the isolation from a wild otter of a multidrug-resistant *M. avium* strain to linezolid, moxifloxacin, streptomycin, isoniazid, trimethoprim/sulfamethoxazole, ciprofloxacin, doxycycline, and ethionamide [60]. The authors discussed the peculiarity of the result, given the general principle that it is not common for wild animals to come into contact with antimicrobials. This finding serves to reinforce the hypothesis that the drug- and multidrug-resistant Mah strains described herein originated from an MSI, as opposed to being driven by drug selective pressure (microevolution) during the course of treatment. However, in order to achieve enhanced strain typing and gain a more profound understanding of the mutations that are responsible for the heteroresistance and diversity in growth abilities documented here, the Mah isolates will be investigated by WGS in a future study.

The treatment of NTM infections is challenging in humans, as well as in animals, owing to the natural drug resistance of some NTMs, the potential emergence of resistance during treatment, and the high incidence of adverse events with unsatisfactory microbiological and clinical outcomes [61]. While no guidelines for the treatment of NTM infections in veterinary medicine have been established, a multidrug therapy based on macrolides, fluoroquinolones, and rifamycins over an extended period is commonly employed. Macrolides (azithromycin and clarithromycin), which inhibit protein synthesis by targeting the bacterial ribosome, are considered the most important component of a treatment regimen for MAC [62]. Within the macrolide class, azithromycin is preferred over clarithromycin due to its superior tolerance and reduced drug interactions (i.e., with rifamycins) [61]. A recent study has proven that the triple combination of clarithromycin, rifabutin, and clofazimine was the most effective in reducing bacterial growth both in an in vitro biofilm assay and in a murine model of *M. avium* subsp. *hominissuis* lung infection [63]. Furthermore, the macrolide azithromycin and clarithromycin exhibited synergistic activity against *Mycobacterium abscessus*, particularly when administered in conjunction with linezolid and cell-wall actin drugs [64].

Rifamycins, which include rifampicin, rifapentine, rifabutin, and rifaximin, constitute a component of the therapeutic regimen employed in the management of tuberculosis (TB) and NTM infections [62]. Rifamycins have been shown to inhibit bacterial RNA polymerase binding to the β subunit [65]. Rifampicin, the most widely utilised rifamycin, possesses high intracellular penetration capacity and exerts a bactericidal effect against both growing and non-growing persistent mycobacteria [66,67]. However, recent studies have called into question its role in the treatment of MAC pulmonary disease (MAC-PD). Pharmacokinetic and pharmacodynamic studies suggest that rifampicin does not enhance the activity of ethambutol and azithromycin in treating *M. avium* infections and does not prevent the emergence of macrolide resistance [68]. This could provide a potential rationale for the suboptimal outcomes and high rates of recurrence observed in cases of MAC-PD [69]. Conversely, rifampicin has been shown to be effective in treating TB, not due to its synergy with other drugs in the treatment regimen (e.g., streptomycin and isoniazide) but rather due to its ability to kill semi-dormant bacilli, which are non-replicating bacteria that undergo occasional metabolic bursts [70]. Pyrazinamide has also been demonstrated to specifically target dormant bacilli [71]. The complexity and dynamicity of TB lesions, together with the different microenvironments generated inside each lesion, induce multiple types of non-replicating bacterial populations with differential drug susceptibility. Consequently, the employment of a combination of drugs has been demonstrated to be highly efficacious in the targeting of these populations across all their locations and in the reduction in the incidence of relapse by addressing multiple bacterial states of infection [70,72].

Fluoroquinolones (e.g., levofloxacin, moxifloxacin, ciprofloxacin) are a group of antibiotics that inhibit the action of two essential enzymes involved in DNA replication and repair: bacterial DNA gyrase and topoisomerase IV. The combination of these antibiotics with other antibiotics has been shown to result in improved treatment outcomes, reduced emergence of resistance, and increased bactericidal activity [61]. In the field of veterinary medicine, the use of clarithromycin and rifampicin in conjunction with fluoroquinolones has been employed in sporadic cases of slowly growing NTM infections in companion animals. As demonstrated in the relevant literature, once mycobacteriosis becomes disseminated, affected animals inevitably succumb or have to be euthanised, regardless of any treatment attempts [4].

In conclusion, we describe a case of a mixed-strain infection of drug and multidrug resistant Mah in a dog with generalised lymphadenomegaly. The long-course, three-based antibiotic treatment proved ineffective in resolving the infection; however, it did enable the dog to survive for a period exceeding four years from the initial diagnosis. Evidence suggests that MAC has the capacity to circulate between the environment and companion animals, and to cause severe infections in its hosts. Given the pathogenetic potential of MAC and the impact of mixed infections caused by this significant group of mycobacteria on companion animals, further investigations are required to achieve a more profound understanding of the prevalence, distribution, and antimicrobial resistance of MAC, as well as of the optimal therapeutic interventions to counteract MSIs with MAC.

## Figures and Tables

**Figure 1 antibiotics-14-00416-f001:**
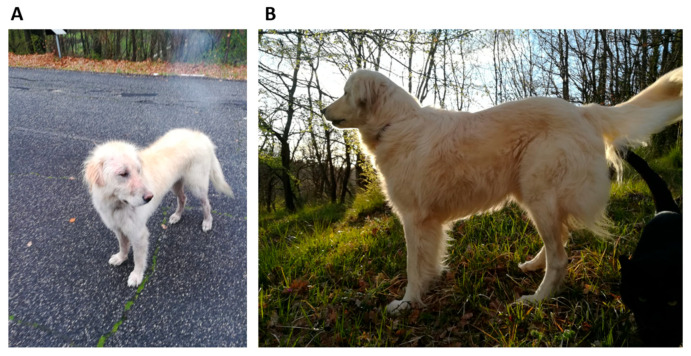
(**A**) The female Maremma sheepdog, aged approximately two years and weighing 19 kg, retrieved from the street. (**B**) The dog six months after the commencement of drug therapy demonstrated a significant improvement in clinical conditions and weight gain.

**Figure 2 antibiotics-14-00416-f002:**
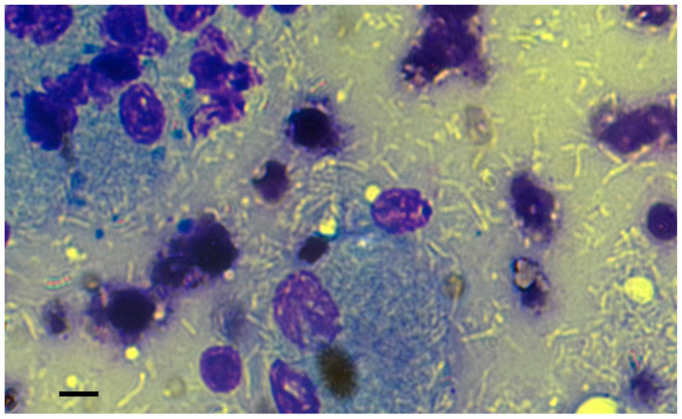
The FNA cytology revealed the presence of numerous negatively stained bacilli, both within and outside the immune cells, lymphocytes, and macrophages. The May–Grünwald–Giemsa stain was utilised for the examination. Bar, 10 µm.

**Figure 3 antibiotics-14-00416-f003:**
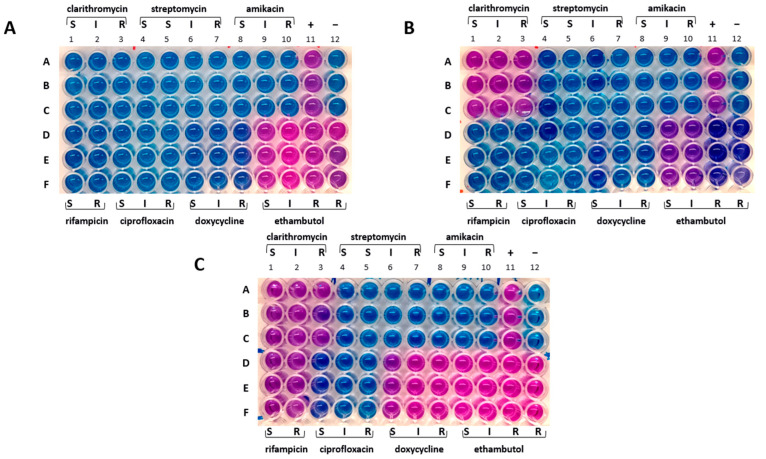
Drug susceptibility profile. (**A**) Mah-Dec2018 pathogen. (**B**) Mah-Apr2021. (**C**) Mah-Nov2022. A–F: sample triplicates. S, susceptible; I, intermediate; R, resistant. Lines A–C, columns 1–3: 16.0 (S), 32.0 (I), and 64.0 (R) µg mL^−1^ for clarithromycin; columns 4–7: 8.0 (S), 16.0 (S), 32.0 (I), and 64.0 (R) µg mL^−1^ for streptomycin; columns 8–10: 16.0 (S), 32.0 (I), and 64.0 (R) µg mL^−1^ for amikacin; column 11: positive control containing no drug (+); column 12: negative control containing uninoculated medium (−). Lines D–F, columns 1–2: 1.0 (S) and 2.0 (R) µg mL^−1^ for rifampicin; columns 3–5: 1.0 (S), 2.0 (I), and 4.0 (R) µg mL^−1^ for ciprofloxacin; columns 6–8: 1.0 (S), 8.0 (I), 16.0 (R) µg mL^−1^ for doxycycline; columns 9–12: 2.0 (S), 4.0 (I), 8.0 (R), 16.0 (R) µg mL^−1^ for ethambutol.

**Figure 4 antibiotics-14-00416-f004:**
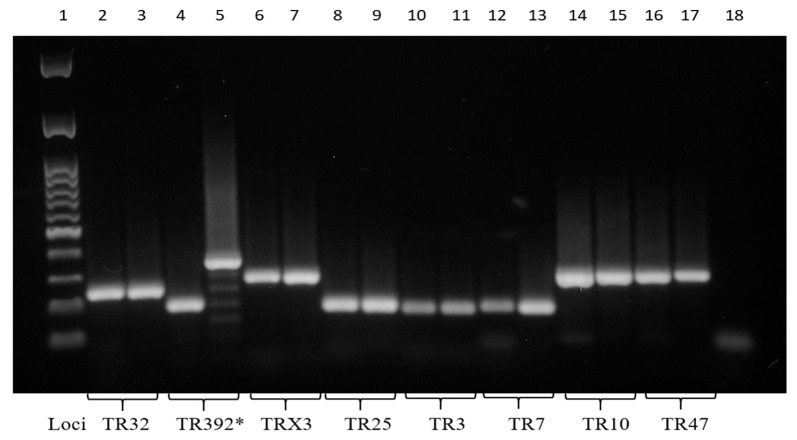
Eight PCR-amplified MIRU-VNTR loci. *, polymorphism at the TR392 locus. The molecular 100 base pairs DNA ladder (column 1); the PCR negative control (column 18). Mah-Dec2018 and Mah-Nov2019 (columns 2, 4, 6, 8, 10, 12, 14, and 16); Mah-Apr2021 (columns 3, 5, 7, 9, 11, 13, 15, and 17). Loci TR32 (columns 2 and 3), TR392 (columns 4 and 5), TRX3 (columns 6 and 7), TR25 (columns 8 and 9), TR3 (columns 10 and 11), TR7 (columns 12 and 13), TR10 (columns 14 and 15), and TR47 (columns 16 and 17).

**Figure 5 antibiotics-14-00416-f005:**
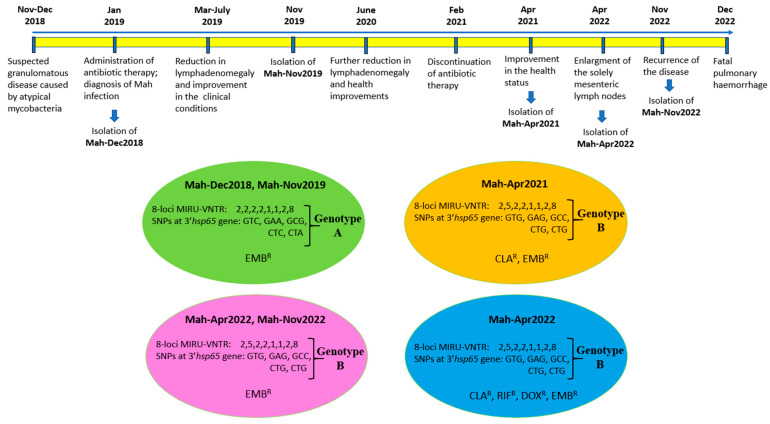
The chronological overview of the most significant events, as well as the various genotypes and phenotypes of Mah isolates. The Mah isolates grouped into four clusters, according to drug susceptibility phenotypes and genotypes. The resistance (^R^) to EMB (ethambutol), CLA (clarithromycin), RIF (rifampicin), and DOX (doxycycline) are shown.

**Table 1 antibiotics-14-00416-t001:** IS-based PCR assays developed for the detection and differentiation of *M. avium* members.

PCR Test	Forward, Reverse Primers	Size (bp)	MAC Member Target *
IS 1311	F 5′-ACTACCGAGAGGAACATCGC-3′ R 5′-CCGTGCAAATAGGCCTCCAT-3′	356	All *M. avium* members
IS 1245	F 5′-TCTGCAAAGACCTCGACA-3′ R 5′-GCATCGGAGATGACCAGTTG-3′	283	Maa, Mah, Mas
IS 901	F 5′-TGAAGGGGTCTGGGATTGG-3′ R 5′-CTACTCCTGTCGTCGCAGTC-3′	208	Maa, Mas
IS 900	F 5′-ACGACTCGACCGCTAATTGA-3′ R 5′-AGCCAGTAAGCAGGATCAGC-3′	656	Map

* Maa, *M. avium* subsp. *avium*; Mah, *M. avium* subsp. *hominissuis*; Mas, *M. avium* subsp. *silvaticum*; Map, *M. avium* subsp. *paratuberculosis*.

**Table 2 antibiotics-14-00416-t002:** IS-based PCR assays developed for the detection and differentiation of MAC members.

**Gene**	**Forward and Reverse Primers**	**Size (bp)**	**Reference**
*gyrB*	F 5′-GCAGACGCCAAAGTCATTGT-3′ R 5′-TCGAACTCGTCGTGAATCCC-3′	353	Armas (2016) [11]
*rpsA*	F 5′-CTTCTCGAATCCCTCGAGCC-3′ R 5′-CGCCTGATCCTGTCCAAGAA-3′	933	Armas (2016) [11]
3′*hsp65*	F 5′-CGGTTCGACAAGGGTTACAT-3′ R 5′-ACGGACTCAGAAGTCCATGC-3′	1059	Turenne (2006) [18]
ITS	F 5′-TTGTACACACCGCCCGTCA-3′ R 5′-TCTCGATGCCAAGGCATCCA-3′	~600	Frothingham (1993) [19]
*rpoB*	F 5′-GCGACACGTCCATGTAGTCC-3′ R 5′-CTGATCAACATCCGTCCCGT-3′	499	This study

**Table 3 antibiotics-14-00416-t003:** Nucleotide changes in the 3′*hsp65* gene.

**Group**	**Codon Position ***				
	**376**	**406**	**423**	**424**	**512**
Mah-Dec2018 Mah-Nov2019	GTC (Val)	GAA (Glu)	GCG (Ala)	CTC (Leu)	CTA (Leu)
Mah-Apr2021 Mah-Apr2022 Mah-Nov2022	GTG (Val)	GAG (Glu)	GCC (Ala)	CTG (Leu)	CTG (Leu)

* The codon position refers to the reference *M. avium* subsp. *hominissuis* MAC 109 strain, for which the complete nucleotide sequence is available in GenBank (sequence ID CP029332.1). The mutated nucleotide is indicated by underlining in relation to the reference sequence. The amino acid coded by the triplet is shown in brackets.

## Data Availability

All relevant data are contained within the article.

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
