# Peer review of "Detection of a Mixed-Strain Infection with Drug- and Multidrug-Resistant Mycobacterium avium Subspecies hominissuis in a Dog with Generalized Lymphadenomegaly"

_antibiotics, 2025, doi:10.3390/antibiotics14040416_

Round 1

Reviewer 1 Report

Comments and Suggestions for Authors

Marianelli et al. attempts to characterize a mixed strain of MAC in a Maremma sheepdog and reports on drug-resistant strains over a long-term study with distinct phenotypes and genotypes of Mah. The findings will be useful for researchers working in the field of veterinary microbiology and expands our understanding of microbial resistance. The manuscript was a pleasure to read and is well written. 

I would like to offer some general suggestions/comments for the authors to improve the mansuscript:

  1. Please consider including a visual timeline of events. There are so many sampling events and therapies over the years of treating this dog. It was difficult to follow.
  2. The dog was treated with a combination of antibiotics. Do these antibiotics have any synergistic effects in the literature? Please elaborate in the discussion. 
  3. The FNA was cultured under aerobic conditions? The oxygenation of (human) lymph nodes is less than 6%, perhaps similar in the dog. What effect does this have on the bacteria suddenly being moved to an aerobic environment? https://tcr.amegroups.org/article/view/17876/html#:~:text=Of%20note%2C%20the%20concentration%20of,and%206%25%20(8). 
  4. The authors should consider adding a section on antibiotic synergism in the discussion. 

Specific comments:

Line 104: Was there a control plate? If so, please describe.

Lines 140-142: The authors should add more details here. How did they account for the number of bacteria/well? How many bacteria were added? In what volume? Many important details are missing.

Line 142: Italicize "in vitro".

Lines 184,216: Can the authors comment on why only the prescapular lymph node was sampled during these times? 

Line 219: Can the authors speculate on this phenomenon? Why did it not grow in broth?

Figure 2. Why were mycoplasma specific stains such as PAS or culture on PPLO agar not performed? It would significantly enhance the manuscript if phenotype of the isolates was included. Were they a mixed colony morphology on the solid plates?

Lines 337-343: Would these drug concentrations be the same in vivo? For example, inside the lymph node?

Author Response

We would like to express our gratitude to the Reviewer for their constructive feedback. Modifications are highlighted in yellow in the text of the revised version of the manuscript.

1. Please consider including a visual timeline of events. There are so many sampling events and therapies over the years of treating this dog. It was difficult to follow.

As suggested, a chronological overview of the most important evens has been shown in Figure 5. Please, see lines 291-292 and the new Figure 5 at page 12.

2. The dog was treated with a combination of antibiotics. Do these antibiotics have any synergistic effects in the literature? Please elaborate in the discussion. 

A thorough discussion was conducted on the therapy administered and the potential synergies between the drugs as outlined in the literature. For further details, please refer to the Discussion section, specifically lines 550-597. The reference list has been updated to reflect this integration.

3. The FNA was cultured under aerobic conditions? The oxygenation of (human) lymph nodes is less than 6%, perhaps similar in the dog. What effect does this have on the bacteria suddenly being moved to an aerobic environment? https://tcr.amegroups.org/article/view/17876/html#:~:text=Of%20note%2C%20the%20concentration%20of,and%206%25%20(8). 

The FNA samples were cultured under aerobic conditions. It was clarified in the text (see line 116).

We would like to thank the reviewer for the interesting point raised, which has offered us with the possibility to explore the impact of oxygen levels on mycobacterial growth. Mycobacteria are classified as obligate aerobes; it’s therefore crucial maintain aerobic conditions within the laboratory setting in order to isolate and culture mycobacteria. Mycobacteria, however, have been shown to be capable of survival in hypoxic conditions (such as those found in the lymph nodes) or anaerobic conditions. In the study conducted by Lewis and Falkinham (see below), non-tuberculous mycobacteria, including M. avium, M. intracellulare and M. scrofulaceum, were examined for their capacity to proliferate under varying oxygen concentrations, such as 21% (air), 12% and 6% (comparable to that found in the lymph nodes), and to survive under conditions of rapid oxygen depletion. The results demonstrated that the growth rates of M. avium in both broth and solid cultures at 21% and 12% oxygen levels were comparable; however, a decrease of 1.4–2.8-fold was observed in the 6% oxygen level. Moreover, both M. avium and the other NTM were found unable to grow under anaerobic conditions. It is therefore crucial to perform cultures under aerobic conditions for optimal isolation of mycobacteria.

Lewis AH, Falkinham JO 3rd. Microaerobic growth and anaerobic survival of Mycobacterium avium, Mycobacterium intracellulare and Mycobacterium scrofulaceum. Int J Mycobacteriol. 2015 Mar;4(1):25-30. doi: 10.1016/j.ijmyco.2014.11.066. Epub 2015 Feb 2. PMID: 26655194.

4. The authors should consider adding a section on antibiotic synergism in the discussion. 

The recommendation made by the reviewer is appreciated. A discussion was conducted on the therapy administered and the potential synergism between the drugs as described in the literature. Please, see lines 491 to 538.

Specific comments:

Line 104: Was there a control plate? If so, please describe.

The cultural isolation of mycobacterial pathogens from potentially infected tissues is generally performed on specific mycobacterial culture media without the preparation of control cultures. This approach is adopted to circumvent cross-contaminations arising from the handling of potentially infected samples and positive controls (cultures of mycobacterial strains) during sample processing. The accuracy and reliability of the microbiological investigations are then assessed by subsequent molecular investigations. This approach was adopted in the present study.

Lines 140-142: The authors should add more details here. How did they account for the number of bacteria/well? How many bacteria were added? In what volume? Many important details are missing.

As suggested, additional information regarding the procedures employed for drug susceptibility testing have been incorporated. Please, see lines 153-159.

Line 142: Italicize "in vitro". Done

Lines 184,216: Can the authors comment on why only the prescapular lymph node was sampled during these times? 

In the case study of lymphadenomegaly presented here, the prescapular lymph nodes were the largest in size and therefore the most readily sampled.

Line 219: Can the authors speculate on this phenomenon? Why did it not grow in broth?

The capacity of bacteria to proliferate in different ways in both solid and liquid media has been documented in the literature. The studies by André E, et al. (2024, Microbiology Spectrum, 12: e0273623) and by Mamila D, et al. (2025, LabMed, 2) found that NTM isolates cultured in agar media exhibited different recovery rates compared to liquid broth cultures. In the present study, we also describe the difference in growth rate of Mah isolates in broth and solid Middlerbrook media. We hypothesise that the composition of the media is a key factor, particularly the protein digest and inorganic salt content, despite minor adjustments to the composition of M7H9 and M7H11 agar. A further potential factor is the oxygen availability, which is known to vary between the microenvironments of broth cultures and agar plates.

Figure 2. Why were mycoplasma specific stains such as PAS or culture on PPLO agar not performed? It would significantly enhance the manuscript if phenotype of the isolates was included. Were they a mixed colony morphology on the solid plates?

We agree with the Reviewer that Mycoplasma may interfere with the isolation and identification of mycobacteria. However, it should be noted that the Middlerbrook M7H9 and M7H11 media utilised in our study are principally employed for the cultivation of mycobacteria and are not suitable for the isolation of Mycoplasma species. Furthermore, we employed molecular-based methodologies (e.g. an IS-based PCR tests and 8-loci MIRU-VNTR assay) as the gold standard for the diagnosis of MAC infection. 

We observed that the primary isolation of mycobacteria from lymph-node-aspirate sample cultures on agar plates might generate colonies that were both translucent and opaque. However, the subsequent molecular typing of these colonies revealed a consistent Mah profile. Our group has previously documented the isolation of M. avium colonies exhibiting mixed morphology in a canine subject afflicted with generalized mycobacteriosis; no association was found between the rpsA polimorphisms and the morphology of the colony (translucent or opaque) (Marianelli et al., 2020, Front. Vet. Sci. 7:569966). These results are consistent with the study of Meylan et al. (Infect Immun. (1990) 58:2564–8. doi: 10.1128/IAI.58.8.2564-2568.1990), which documented the generation of both opaque and transparent colonial variants when MAC strains were cultivated in vitro. Therefore, colony morphology observed in vitro can reflect the stress growth conditions a microorganism experiences, as previously documented (Chantratita N, et al. J Bacteriol. 2007 Feb;189(3):807-17. doi: 10.1128/JB.01258-06).

Lines 337-343: Would these drug concentrations be the same in vivo? For example, inside the lymph node?

The concentration of drugs that can be achieved within a single tissue or lymph node remains unknown. The selection of drugs and combinations of drugs, as well as the dosage, are critical issues in the treatment of mycobacterial infections. Mycobacterial infections present with a wide range of lesions, each of which consists of multiple microenvironmental locations (Greenstain and Aldridge, Front Cell Infect Microbiol. 2023 Jan 6;12:1085946). The heterogeneity of the lesion structure affects the drug access, absorption and metabolism. MALDI mass spectrometry imaging has demonstrated that drugs access differently into the different niches or locations of a lesion in vivo (Sarathy et al., ACS Infect Dis 2016, 2:552-563). Furthermore, pharmacokinetic modelling studies have demonstrated that drugs are able to accumulate differently inside lesion compartments. Consequently, modelling of lesion drug access may facilitate prediction of effective drug combinations capable of reaching diverse locations and types of drug-tolerant and drug-sensitive bacteria (Greenstrain and Aldridge, 2023).

Reviewer 2 Report

Comments and Suggestions for Authors

1. In the abstract, it could be beneficial to briefly emphasize the clinical implications more explicitly (e.g., difficulty in treating Mah due to emerging resistance during therapy).

2. I suggest author to briefly highlight gaps in veterinary knowledge about mixed infections in Introduction section. State why documenting such infections in animals is important, and articulate the potential implications for disease management and public health.

3. The current narrative format makes it challenging for readers to follow the logical sequence of analyses performed. To improve clarity, the methodology should ideally be divided into clearly titled subsections. Adding these subsections would significantly enhance readability and make it easier for readers to quickly identify and understand each methodological step.

4. Although the authors interpret the presence of different Mah genotypes as indicative of mixed infections, discussing more explicitly alternative explanations (like laboratory contamination or methodological errors) and how these were ruled out would further strengthen credibility.

5. Please add more statement for considering whole genome sequencing (WGS) to improve strain resolution and accurately map resistance mutations in future work.

6. As the authors rightly state, standard molecular techniques may fail to detect less dominant strains in mixed populations, give more emphasizing statement for deep sequencing or metagenomic tools to overcome this diagnostic blind spot.

7. Please add more discussion that compares existing literature on Mah in dogs and other animals and this study. More explicit comparison of its findings (e.g., strain profiles, resistance patterns, treatment response) could be discussed.

Comments on the Quality of English Language

Please proof read again the manuscript, especially the introduction section.

Author Response

We would like to express our gratitude to the Reviewer for their constructive feedback. Modifications are highlighted in light blue in the text of the revised version of the manuscript.

  1. In the abstract, it could be beneficial to briefly emphasize the clinical implications more explicitly (e.g., difficulty in treating Mah due to emerging resistance during therapy).

We did not document resistance to the three drug-based therapy during the course of treatment. Indeed, from early January 2019 to late February 2021(the period of therapy), the Mah isolates (i.e. Mah-Dec2018 and Mah-Nov2019) demonstrated sensitivity to the therapeutic regimen comprising rifampicin, clarithromycin and ciprofloxacin; the treatment plan was subsequently validated. Conversely, the interruption of the drug therapy was followed by the emergence of drug resistance.

  1. I suggest author to briefly highlight gaps in veterinary knowledge about mixed infections in Introduction section. State why documenting such infections in animals is important, and articulate the potential implications for disease management and public health.

As suggested, the issue of mixed infections in veterinary medicine has been briefly outlined in Introduction. For further details, please refer to lines 64-73.

  1. The current narrative format makes it challenging for readers to follow the logical sequence of analyses performed. To improve clarity, the methodology should ideally be divided into clearly titled subsections. Adding these subsections would significantly enhance readability and make it easier for readers to quickly identify and understand each methodological step.

We would like to express our gratitude to the Reviewer for the valuable suggestion, which has significantly enhanced the readability of the manuscript. The addition of subsections and a graph illustrating the chronology of the most salient events (new Figure 5) have been incorporated into the manuscript.

  1. Although the authors interpret the presence of different Mah genotypes as indicative of mixed infections, discussing more explicitly alternative explanations (like laboratory contamination or methodological errors) and how these were ruled out would further strengthen credibility.

We exclude a laboratory cross-contamination as both the SNPs and heteroresistance documented here have never previously been identified in our laboratory. Furthermore, in order to verify the accuracy of the results, the DNA extraction, PCR amplifications and nucleotide sequencing processes were repeated on two separate occasions, independently. In contrast, the drug susceptibility testing results were confirmed by performing the tests on three separate occasions. The results obtained were reproducible. This has been better clarified in the text. For further details, please refer to lines 166 and 317.

  1. Please add more statement for considering whole genome sequencing (WGS) to improve strain resolution and accurately map resistance mutations in future work.

We are aware of the high level of resolution of WSG and we intend to undertake a more comprehensive study of these Mah isolates using WGS in the near future. We stated that in the manuscript. Please, see lines 546-549.

  1. As the authors rightly state, standard molecular techniques may fail to detect less dominant strains in mixed populations, give more emphasizing statement for deep sequencing or metagenomic tools to overcome this diagnostic blind spot.

The use of WGS and metagenomics has been emphasised in the discussion section. For further details, please refer to lines 453-463.

  1. Please add more discussion that compares existing literature on Mah in dogs and other animals and this study. More explicit comparison of its findings (e.g., strain profiles, resistance patterns, treatment response) could be discussed.

A thorough discussion was conducted on MSIs in other animals as outlined in the literature. For further details, please refer to the Discussion section, specifically lines 427-442. The reference list has been updated to reflect this integration.

Comments on the Quality of English Language

Please proof read again the manuscript, especially the introduction section.

We would like to express our gratitude for the suggestion. The entire manuscript has been meticulously proofread to rectify any errors.

Reviewer 3 Report

Comments and Suggestions for Authors

The manuscript describes a case report of a mixed strain infection of Mycobacterium in a dog who presented with generalised lymphadenopathy. The authors have discussed the case in details and provided all the relevant information pertaining to case presentation, diagnosis, cultures, management as well as outcomes.

My only suggestion is to cut short the text under case description; the authors may add a flowchart depicting the sequence of events in chronology to make it more interesting to the readers.  

Comments on the Quality of English Language

Minor editing in English language required.

Author Response

We would like to express our gratitude to the Reviewer for the valuable suggestion, which has significantly enhanced the readability of the manuscript. Subsections and a graph illustrating the chronology of the most salient events (new Figure 5) have been incorporated into the manuscript.